# Differential Effects of Platelet Factor 4 (CXCL4) and Its Non-Allelic Variant (CXCL4L1) on Cultured Human Vascular Smooth Muscle Cells

**DOI:** 10.3390/ijms23020580

**Published:** 2022-01-06

**Authors:** Dawid M. Kaczor, Rafael Kramann, Tilman M. Hackeng, Leon J. Schurgers, Rory R. Koenen

**Affiliations:** 1Department of Biochemistry, Cardiovascular Research Institute Maastricht (CARIM), Maastricht University, P.O. Box 616, 6200 MD Maastricht, The Netherlands; d.kaczor@maastrichtuniversity.nl (D.M.K.); t.hackeng@maastrichtuniversity.nl (T.M.H.); l.schurgers@maastrichtuniversity.nl (L.J.S.); 2Institute of Experimental Medicine and Systems Biology, RWTH Aachen University, Pauwelsstrasse 30, 52074 Aachen, Germany; rkramann@ukaachen.de; 3Division of Nephrology and Clinical Immunology and Medical Faculty, RWTH Aachen University, Pauwelsstrasse 30, 52074 Aachen, Germany; 4Erasmus Medical Center, Department of Internal Medicine, Nephrology and Transplantation, Doctor Molewaterplein 40, 3015 GD Rotterdam, The Netherlands

**Keywords:** CXCL4, CXCL4L1, smooth muscle cell, inflammation, artery, vascular remodeling, platelet

## Abstract

Platelet factor 4 (CXCL4) is a chemokine abundantly stored in platelets. Upon injury and during atherosclerosis, CXCL4 is transported through the vessel wall where it modulates the function of vascular smooth muscle cells (VSMCs) by affecting proliferation, migration, gene expression and cytokine release. Variant CXCL4L1 is distinct from CXCL4 in function and expression pattern, despite a minor three-amino acid difference. Here, the effects of CXCL4 and CXCL4L1 on the phenotype and function of human VSMCs were compared in vitro. VSMCs were found to constitutively express CXCL4L1 and only exogenously added CXCL4 was internalized by VSMCs. Pre-treatment with heparin completely blocked CXCL4 uptake. A role of the putative CXCL4 receptors CXCR3 and DARC in endocytosis was excluded, but LDL receptor family members appeared to be involved in the uptake of CXCL4. Incubation of VSMCs with both CXCL4 and CXCL4L1 resulted in decreased expression of contractile marker genes and increased mRNA levels of KLF4 and NLRP3 transcription factors, yet only CXCL4 stimulated proliferation and calcification of VSMCs. In conclusion, CXCL4 and CXCL4L1 both modulate gene expression, yet only CXCL4 increases the division rate and formation of calcium-phosphate crystals in VSMCs. CXCL4 and CXCL4L1 may play distinct roles during vascular remodeling in which CXCL4 induces proliferation and calcification while endogenously expressed CXCL4L1 governs cellular homeostasis. The latter notion remains a subject for future investigation.

## 1. Introduction

Platelet factor 4 (PF4 or CXCL4) is a 7.8 kDa chemokine, synthesized mainly by megakaryocytes and stored in the α-granules of platelets. Unlike most chemokines, there is no single receptor that mediates the actions of CXCL4. Instead, CXCL4 can bind to different receptors, initiating various downstream signaling pathways on distinct cell types [1]. Despite the identification of several putative receptors e.g., glycosaminoglycans (GAGs) [2,3], CXCR3 [4,5,6], CCR1 [7], and low-density lipoprotein (LDL) family receptors [8,9], the molecular mechanisms behind the biological functions of CXCL4 remain incompletely clarified [10].

Upon injury or during the formation of an atherosclerotic plaque, CXCL4 was found to be transported from the blood into deeper layers of the vessel [11]. Over the past 40 years, studies in mice and humans have confirmed the presence of CXCL4 (and not platelets) in the atherosclerotic plaques [12,13,14].

Recently, CXCL4 has received major attention due to its involvement in the pathophysiology of vaccine-induced thrombocytopenia, a rare and severe complication after vaccination against SARS-CoV-2 [15,16].

Vascular smooth muscle cells (VSMCs) in the arterial vessel wall are considered heterogeneous in phenotype and display a high degree of plasticity. In the vascular media, the contractile VSMC phenotype is important in maintaining tissue elasticity, wall homeostasis and vascular tone [17,18]. During pathologic vascular remodeling processes, VSMCs migrate into the intima of the vessel wall and undergo a phenotypic switch towards a synthetic state, hallmarked by dedifferentiation with reduced or absent expression of VSMC-specific genes, increased production of extracellular matrix components and inflammatory cytokines [19]. Interestingly, CXCL4 also contributes to the phenotypic switch of VSMCs towards synthetic states [20,21]. Previous work showed that CXCL4 treatment of VSMCs increased migration and proliferation and decreased the level of contractile markers (calponin, α-actin), consistent with a synthetic phenotype [20]. CXCL4 also co-localizes with calcified regions of human carotid atherosclerotic plaques [12]. Moreover, during plaque development, CXCL4 contributes to the formation of VSMC-derived foam cells, by enhancing the binding of oxidized low-density lipoprotein to hVSMCs [22].

CXCL4L1 is a non-allelic variant of CXCL4 that has a different leader peptide and 3 amino acid substitutions at the C-terminus. It is also stored in platelets and released after activation [23,24,25]. Until now, the knowledge of CXCL4 biology is mainly based on studies on native CXCL4, whereas CXCL4L1 has not been studied to this extent. Despite small differences in the primary structure, CXCL4L1 is highly distinct from CXCL4 with regards to 3D-structure and function [26,27]. The two chemokines share several properties such as anti-angiogenic and anti-tumor effects in vivo, when exogenously added as isolated proteins [28], but major differences also exist, such as in binding to GAGs, export, and interaction with receptors or oligomerization [27]. The expression and release of CXCL4 and CXCL4L1 is different in various cell types. In contrast to CXCL4L1, which is continuously synthesized and secreted through a constitutive pathway, CXCL4 is stored in secretory granules and released upon activation (i.e., in platelets) [24]. While CXCL4 mainly has proinflammatory effects in atherogenesis, CXCL4L1 may rather exert homeostatic roles [1,29]. This suggests that the roles of CXCL4 and CXCL4L1 in homeostatic and inflammatory processes are different [24]. Given the versatile role of platelets in atherosclerosis, vascular remodeling and immune regulation, we aimed at elucidating possible functional differences between CXCL4 and CXCL4L1 more precisely, with particular focus on their effects on VSMC function in relation to vascular wall remodeling.

## 2. Results

### 2.1. Platelet Factor 4 (CXCL4), but Not CXCL4L1 Is Internalized by Vascular Smooth Muscle Cells (VSMCs)

Previous work has demonstrated rapid internalization of CXCL4 by endothelial cells [30]. To investigate whether CXCL4 and its variant CXCL4L1 are taken up by VSMCs, cells were treated with CXCL4 or CXCL4L1, stained with fluorescent antibodies and the measured fluorescence was quantified. The analysis revealed that the internalization of CXCL4 by VSMCs is significantly increased in comparison with control cells at 37 °C (Figure 1A). Interestingly, a baseline level of intracellular CXCL4L1 expression appeared to be present in VSMC, as visualized by staining using a CXCL4L1-specific antibody (Figure 1B). Intracellular CXCL4 levels were further increased by the addition of CXCL4, but not by CXCL4L1 at 37 °C (Figure 1A,C). After addition, CXCL4 staining appeared as a speckled pattern, suggesting endosomal localization. This was visible also without permeabilization, after removal of surface-bound CXCL4 using heparin. At 4 °C, no CXCL4 uptake was observed (Appendix A).

The results obtained suggest that CXCL4L1 is constitutively expressed and that CXCL4 is internalized by VSMCs, which occurs by an energy-requiring process at physiological temperatures.

### 2.2. The Uptake of CXCL4 into VSMCs Is Mediated by the Low-Density Lipoprotein (LDL) Receptor Family

To investigate possible surface receptors that might be involved in a CXCL4 internalization by VSMCs, the candidates CXCR3, DARC (atypical chemokine receptor 1, ACKR1) and the LDL-receptor family were tested. After confirming the presence of CXCR3 and DARC on VSMCs (Appendix A), their possible role in the uptake of CXCL4 and CXCL4L1 by VSMCs was investigated in the presence of blocking CXCR3 or DARC antibodies. Although a significantly increased uptake of CXCL4 was demonstrated, this was not affected by a blockade of CXCR3 or DARC (Figure 2A). This suggests that these receptors do not play a role in CXCL4 endocytosis by VSMCs.

Next, endocytosis was blocked by the clathrin- and dynamin inhibitors Pitstop2 and Dynasore, respectively, to investigate whether the uptake of CXCL4 occurred by endocytosis. While the uptake of fluorescently labelled native LDL particles (DiI-nLDL) was effectively blocked by both Pitstop2 and Dynasore (Appendix A), no effect of these endocytosis inhibitors on the internalization of CXCL4 was observed (Figure 2B). These data suggest that receptor-mediated endocytosis is not involved in CXCL4 uptake to VSMCs.

To further explore the mechanism of CXCL4 uptake, LDL receptor family members were investigated [20]. The addition of RAP, a well-established blocker of these receptors, led to a significantly decreased endocytosis of DiI-nLDL particles by VSMCs (Appendix A) and significantly diminished CXCL4 uptake by VSMCs (Figure 2C). Similar to the other results, there was virtually no CXCL4L1 uptake by VSMCs under any condition (Figure 2 and Appendix A). In previous studies, CXCL4 was found to interact with oxidized LDL (oxLDL) [22]. However, co-incubation of CXCL4 or CXCL4L1 with DiI-oxLDL did not affect the internalization of DiI-oxLDL by VSMCs, compared to control (Appendix A). 

### 2.3. CXCL4 Is Not Internalized by Heparin Pre-Treated Quiescent VSMCs

Human VSMCs are characterized by high phenotypic plasticity. To address the question of whether uptake of CXCL4 and CXCL4L1 differs by phenotype of VSMCs, contractile or synthetic phenotypes were induced by pre-treatment with heparin or PDGF, respectively. There was a significant uptake of CXCL4 and not CXCL4L1 in control cells and PDGF-pretreated cells (synthetic phenotype) (Figure 3A,B). Markedly, while induction of a synthetic phenotype with PDGF did not further enhance CXCL4 uptake, pre-treatment with heparin, followed by washing and overnight starvation completely blocked the internalization of CXCL4 by VSMCs (Figure 3A,B). The effect of heparin appeared to be specific since induction of a contractile phenotype by culturing the cells in low-serum medium did not reduce CXCL4 uptake (Appendix A). The data suggest that in this experimental setup, CXCL4 internalization is specifically blocked by heparin pre-treatment and might not be a general feature of contractile VSMCs. Of note, since heparin can neutralize CXCL4, it was removed >12 h before addition of the chemokines.

### 2.4. CXCL4 Modifies Expression of Phenotype Switch-Specific Genes in VSMCs

To investigate whether CXCL4 and CXCL4L1 treatment affect the phenotype of VSMCs on a molecular level, qPCR analysis was performed. We made a selection of contractile markers (calponin—CNN1, alpha smooth muscle actin—αSMA) and transcription factors (Krüppel-like factor 4—KLF4, NLR family pyrin domain containing 3—NLRP3). The cells were incubated with the chemokines for 24 or 72 h, and cDNA was subjected to RT-qPCR. The data show that mRNA levels of CNN1 and α-SMA were significantly lower after 24 h treatment with CXCL4 and CXCL4L1 in comparison to control (Figure 4A,B). On the other hand, although accompanied by a notable experimental variation, mRNA levels of the transcription factors KLF4 and NLRP3 were significantly higher after 24-h incubation with CXCL4 and CXCL4L1 (Figure 4C,D). Interestingly, after 72 h of incubation with CXCL4 or CXCL4L1 all markers remained unchanged and comparable to controls (Appendix A), indicating a transient influence on gene expression.

### 2.5. Effects of CXCL4 and CXCL4L1 on Inflammatory and Calcification Potential of VSMCs

Consistent with previous studies [20,21], a stimulatory and dose-dependent effect of CXCL4 on proliferation of VSMCs was observed, when compared to control (Figure 5A,B). Interestingly, CXCL4L1 did not affect proliferation of VSMCs at any concentration (Figure 5A,B).

To examine whether the incubation of VSMCs with CXCL4 or CXCL4L1 stimulates ROS formation, a fluorogenic dye-based ROS assay was performed. Although ROS formation was observed after treatment with hydrogen peroxide (positive control), treatment with CXCL4 or CXCL4L1 did not result in increased ROS production after 6 h of incubation (Figure 6A). Interestingly, there was virtually no change in ROS levels after the treatment with CXCL4 or CXCL4L1 with or without calcium-phosphate supplementation (Figure 6A).

To investigate whether CXCL4 and CXCL4L1 play a role in the formation of calcium-phosphate deposits on the cell surface, a calcification assay was performed. The cells were treated with calcium-phosphate alone or combined with CXCL4 or CXCL4L1 at 1 or 10 µg/mL. The addition of CXCL4 at 1 µg/mL did not significantly alter calcification but CXCL4 at 10 µg/mL increased calcification of cultured VSMCs by approximately 50% (Figure 6B). Notably, CXCL4L1 at lower and higher concentration did not significantly affect the formation of calcium-phosphate deposits on the cell surface, with a tendency to lower calcification (Figure 6B).

## 3. Discussion

In this study, we demonstrated that CXCL4L1 and CXCL4 play distinct roles in the phenotypic modulation of VSMCs. Smooth muscle cells have been shown to play an important role in vascular remodeling, atherosclerosis and vascular calcification [19,31]. Previous studies have implicated CXCL4 in the induction of inflammation and phenotype change in VSMCs [20,21], but the role of its non-allelic variant (CXCL4L1) has been unclear. Messenger RNAs of both variants of platelet factor 4 were found in VSMCs and other cell types, but VSMCs were the only cell type showing a predominance of CXCL4L1 expression over the native CXCL4 [24]. By staining with a specific antibody, we confirmed these observations on the cellular level, showing that non-treated VSMCs have a baseline content of endogenous CXCL4L1 protein. Since CXCL4 is internalized by endothelial cells [30], we investigated whether CXCL4 is endocytosed by VSMCs or rather binds to the cell surface, resulting in the intracellular signaling without concurrent endocytosis. Our data clearly indicate that unlike its variant, exogenously added, native CXCL4 was internalized by VSMCs and appeared to be enclosed within endosome-like structures.

It is well known that chemokines and chemokine receptors form a redundant system. One chemokine can bind to different receptors and vice versa. Recent evidence suggests that CXCR3 (CXCR3A, -B and -alt) could serve as a functional receptor for both CXCL4 and CXCL4L1 [5,6,32,33,34] and DARC for native CXCL4 [35]. Although we found the presence of both receptors in cultured VSMCs, their blockade by specific antibodies did not affect the internalization of CXCL4. This is contrary to the aforementioned studies [4,6,33,35], but in accordance with our previous observation that CXCR3 was not involved in CXCL4 uptake in endothelial cells [30]. In another report, a CXCR3 blocking antibody did not affect CXCL4-induced cytokine production in the human carotid artery SMCs [20].

The most common route of endocytosis of chemokine receptors involves clathrin and during membrane budding and scission also requires the enzyme dynamin [36]. Here, we proved that the endocytosis of CXCL4 by VSMCs was clathrin- and dynamin-independent. This contrasts with our previous work showing an involvement of both proteins during endocytosis of CXCL4 by endothelial cells [30] suggesting distinct uptake mechanisms in different cell types. Alternatively, chemokines were shown to be internalized through clathrin-independent mechanisms involving lipid-rafts and caveolae [37]. The involvement of caveolae-mediated endocytosis in CXCL4 internalization is less likely since this process also requires dynamin for membrane scission [38]. On the other hand, endocytosis of rafts, characterized by cholesterol sensitivity may include caveolin- and dynamin-dependent or independent pathways [39]. Therefore, the exact CXCL4 endocytosis route in VSMCs needs to be investigated more precisely.

Previously CXCL4 was reported to signal through low-density lipoprotein receptor-related protein-1 (LRP1) [9] and LDL receptor (LDLR) [8] in endothelial cells. Here, we showed that the uptake of CXCL4 was abrogated after treatment with a universal inhibitor of the LDL receptor family (RAP). Since RAP blocks all members of the LDLR family, further studies using specific blocking antibodies or RNA interference are required to pinpoint a specific receptor for CXCL4. Taken together, our findings indicate that CXCL4 uptake in hVSMCs is mediated by the LDL receptor family and may signal mainly through LRP1, which was previously reported by Shi et al. [20]. In this respect it might appear paradoxical that, while sharing the receptors, LDL enters the cell by receptor-mediated endocytosis, whereas CXCL4 is taken up independently of clathrin or dynamin. This also contrasts with our previous observations of CXCL4 uptake in endothelial cells, which did appear to occur by clathrin- and dynamin-dependent endocytosis. The LDL-receptor family members might serve as a functional docking site for CXCL4, while entry into the cell is mediated by a different process. 

The interactions of CXCL4 and oxidized LDL (oxLDL) was also investigated. Oxidized LDL can bind to the LOX-1 receptor on VSMCs and upregulate expression of several cytokines (e.g., IL-1β and TNF-α), which may result in a proinflammatory phenotype [40]. Former observations suggested that CXCL4 promoted binding of oxLDL to VSMCs and enhanced the internalization of oxLDL [22]. In this study we showed that the co-addition of oxLDL with CXCL4 and CXCL4L1 did not affect endocytosis of oxLDL.

Since the phenotype might influence the uptake of CXCL4, VSMCs were treated with PDGF-BB or heparin to direct the cells towards synthetic or contractile phenotypes, respectively [41,42,43]. We hypothesized that the different composition of extracellular matrix proteins and GAGs (CXCL4 co-receptors) on various VSMCs phenotypes can affect the internalization of investigated chemokines. Unlike PDGF-induced synthetic cells, heparin-induced contractile cells did not take up CXCL4. This effect was not observed when low-serum was used to induce a contractile phenotype. An explanation of the action of heparin on vascular cells is the removal of various growth factors attached to GAGs, decrease of ERK signaling or inhibition of proliferation and transcription factor activity in VSMCs [44]. Taking this into account, we conclude that the blockade of CXCL4 internalization was caused by a heparin-specific phenotypic switch of VSMCs and might not be a general feature of contractile VSMCs. 

Treatment of VSMCs with CXCL4 and CXCL4L1 resulted in decreased expression of both CNN-1 and ACTA-2, indicating a phenotype switch from contractile to synthetic. These data confirmed the previous findings of Shi et al. [20]. The addition of CXCL4 and CXCL4L1 also elevated the expression of Krüppel-like factor 4 (KLF4), a transcription factor that directs cell fate, differentiation and proinflammatory response of VSMCs [45,46] and was previously found to be upregulated in VSMCs during atherogenesis [47] and upon vascular injury [46]. This is also in line with previous observations that CXCL4 accelerated VSMCs inflammatory responses to injury, in part by stimulating the expression of KLF4 [20]. The activation of NLRP3 inflammasome has been previously shown to contribute to VSMCs phenotypic transformation and proliferation in rats and primary VSMCs [48]. To the best of our knowledge, the effect of CXCL4 and CXCL4L1 on the NLRP3 mRNA stimulation has not been previously described. In a recent publication, soluble platelet factors were found to enhance NLRP3 transcription and boost IL-1 expression in macrophages [49]. Physiological concentrations of CXCL4 were shown to directly stimulate human VSMCs inflammatory responses, with partly KLF4-dependent increase of IL-6 and CXCL8 secretion [20]. CXCL4 has also been shown to boost IL-6 and TNF-α production in monocytes [50,51] and bone marrow stroma [52]. In contrast, intrahepatic mRNA IL-6 expression was significantly increased in CXCL4 knockout mice compared to wild-type mice [53]. It is interesting that both CXCL4 and CXCL4L1 induced a change in the expression of the above markers in VSMCs. This indicates that internalization is not strictly required for these effects, since only CXCL4 was shown to be taken up and both chemokines might activate VSMCs through their cognate surface receptors.

Already in the 1970s it was suggested that platelets have a causative role in VSMCs proliferation and CXCL4 was implied among the mitogens carried by platelets. In this study, we showed that CXCL4 stimulated the proliferation of VSMCs, which is in line with previous observations [20,21,54]. These results indicate a role for CXCL4 in the induction of intimal thickening following vascular injury through direct or indirect pathways and subsequent induction of medial SMCs proliferation and migration. Interestingly, it was previously reported that both variants inhibited the proliferation of endothelial cells [34,55] and in this study CXCL4L1 did not affect VSMC proliferation. Therefore, the baseline expression of CXCL4L1 in VSMCs might represent a physiologic regulatory mechanism to maintain vascular homeostasis and to avoid endothelial cell proliferation and aberrant angiogenesis [24].

Oxidative stress also participates in pathophysiological processes of vascular damage. It has been shown that CXCL4 induced ROS production in various cell types [56,57] and that the CXCL4L1 did not affect ROS generation in rat retinas [58]. Recently it was also demonstrated that ROS production in VSMCs is required for the release of extracellular vesicles that promote calcification [59]. However, in our experimental setup neither CXCL4 nor CXCL4L1 was able to stimulate ROS production in VSMCs.

We also hypothesized that CXCL4 may facilitate the formation of microcalcifications that leads to plaque destabilization and rupture. Previous studies in mouse and human revealed that CXCL4 was present in atherosclerotic plaques as well as regions of calcification [12,14] and its elimination from platelets reduced atherosclerosis [14,60]. Here, we showed that although the experimental variation was large, CXCL4 appeared to increase calcification at higher concentrations while the addition of CXCL4L1 showed a non-significant trend towards a reduction. Unlike CXCL4, CXCL4L1 did not bind to CCL5 and did not enhance atherosclerosis in transgenic mice with CXCL4 modified to CXCL4L1, indicating that CXCL4L1 has reduced inflammatory potential, at least during plaque development [29]. On the other hand, a recent study showed that CXCL4L1 induced a different palette of cytokines and chemokine receptors in monocytes than CXCL4, skewing them towards a pro-inflammatory phenotype [61]. Thus, the question of whether CXCL4L1 exerts a homeostatic rather than an inflammatory function can likely only be answered within a defined cellular context. To further study the role of endogenous CXCL4L1, its expression can be downregulated by RNA interference. However, this might be accompanied by phenotypic and expression changes induced by liposome- or viral-mediated transfection methods. Alternatively, the CXCL4L1 gene can be excised using CRISPR/CAS9 technology in induced pluripotent stem cells, prior to phenotypic induction to SMC.

Taken together, CXCL4 is taken up into SMC by the action of the LRP receptor (or LDL-receptor family). The receptors CXCR3 and LRP might mediate the signaling by CXCL4. CXCL4L1, on the other hand, is not internalized by SMC, but expressed endogenously. Previous studies have suggested CXCR3 as a functional receptor for CXCL4L1 [6]. This is schematically summarized in Figure 7. This study provides further evidence for the pro-inflammatory actions of CXCL4 on vascular cells and highlights the distinct properties of the highly homologous chemokines CXCL4 and CXCL4L1. 

## 4. Materials and Methods

### 4.1. Reagents

CXCL4 was isolated from anonymous expired platelet packs as described elsewhere [23]. Recombinant CXCL4L1 was expressed in *E. coli* and purified in our laboratory as described previously [26,55]. Products were analyzed by Xevo^®^ UPLC-MS (Waters Corporation, Milford, MA, USA) as described and results are shown in Appendix A [62]. Receptor-associated protein (RAP) was purchased from Enzo Life Sciences (Lörrach, Germany). Native human LDL (SAE0053), Dynasore (CAS: 304448-55-3) and PitStop2 (CAS: 1419093-54-1) and TRI reagent were purchased from Merck Millipore (Darmstadt, Germany). Interleukin 6 (IL) ELISA kits, 1,1′-dioctadecyl-3,3,3′,3′-tetramethylindocarbocyanine perchlorate (DiI)-labeled oxLDL, 2’,7′-dichlorofluorescin diacetate (DCFDA), and attachment factor (AF), anti-DARC clone 2C3, polyclonal anti-CXCL4L1 antibody, and goat anti-rabbit antibodies conjugated with AF532 were purchased from ThermoFisher Scientific (Waltham, MA, USA). Cytokines and rabbit anti-human CXCL4 were from Peprotech (Rocky Hill, NJ, USA). Monoclonal mouse anti-CXCR3 clone # 49801 and monoclonal mouse anti-DARC, clone #358307 were from R&D systems (Minneapolis, MN, USA). FITC-conjugated goat anti-mouse from Jackson ImmunoResearch (Ely, UK). MicroBCA kit and iScript^TM^ Reverse Transcription Supermix were from Bio-Rad (Hercules, CA, USA). Takyon™ No Rox SYBR® MasterMix dTTP blue was from Eurogentec (Seraing, Belgium). O-cresolphthalein assay was purchased from Randox (Crumlin, UK).

### 4.2. Cell Culture

Human vascular smooth muscle cells (VSMCs) were isolated from a single patient in our laboratory from tissue explants (human thoracic aorta) as described previously and cultured between passage 4 and 10 [59,63]. Collection, storage, and use of tissue and human aortic samples were performed in agreement with the Dutch Code for Proper Secondary Use of Human Tissue.

### 4.3. CXCL4/CXCL4L1 Internalization

VSMC were seeded in attachment factor-coated black 96-well plate at a density of 3–5 × 10^3^ cells/well for 16–20 h and subsequently starved in DMEM with 0.5% FCS overnight. After that, cells were incubated with the vehicle (50 mM Na-Acetate pH 5.5, 600 mM NaCl and 0.5 mM EDTA), 0.5 µg/mL CXCL4 or CXCL4L1 in DMEM containing 0.5% FCS or 0.3% BSA (for oxLDL-experiments) for 1 h, either at 37 °C or 4 °C.

In some experiments, blocking antibodies (10 µg/mL of anti-CXCR3 or anti-DARC), Dynasore (80 µM) or PitStop2 (15 µM), (DiI)-labeled oxLDL (5 µg/mL), RAP (200 nM) or the vehicle, were added for up to 30 min, and then 0.5 µg/mL of the chemokines (CXCL4 or CXCL4L1) were added to the cells and the culture plates were incubated for additional 1 h at 37 °C.

For experiments on VSMC phenotype, cells were pre-incubated without or with PDGF (20 ng/mL, 3 days) or heparin (200 U/mL, 5 days) in DMEM with 20% FCS, or cultured in low serum DMEM (0.5% FCS). After up to 5 days, cells were starved in DMEM with 0.5% FCS overnight before addition of the chemokines. 

Then, the cells were washed (200 U/mL heparin in PBS) to remove remaining surface chemokines. After that, cells were fixed with 4% paraformaldehyde and blocked with PBS containing 2% BSA with or without 0.1% Triton X-100 for 1 h at room temperature. The cells were stained as described below.

### 4.4. LDL Receptor Family Blockade by RAP or Endocytosis Inhibitors

VSMCs were cultured as above, starved in DMEM/0.3% BSA overnight, and the cells were pre-treated with vehicle or RAP (200 nM), Dynasore (80 µM) or PitStop2 (15 µM) for 15–30 min. Subsequently, the cells were incubated with the vehicle, DiI-nLDL (5 µg/mL) alone or RAP and DiI-nLDL together for an additional 1 h at 37 °C. Afterwards, the cells were incubated with 200 U/mL heparin for 5 min. Subsequently, cells were fixed with 4% paraformaldehyde for 10 min at room temperature, washed with PBS, stained with Hoechst and analyzed with Cytation^TM^ (BioTek, Agilent, Santa Clara, CA, USA).

### 4.5. Immunocytochemistry

The primary antibodies (rabbit anti-human CXCL4(L1), monoclonal mouse anti-CXCR3, monoclonal mouse anti-ACKR1/DARC) were added to VSMCs at a final concentration of 2 µg/mL or 10 µg/mL (CXCR3, DARC) and incubated overnight at 4 °C. After washing, the secondary antibody was added (goat anti-rabbit AF532, goat anti-mouse-FITC) at a final concentration of 5 µg/mL for 1 h. Next, cells were washed, and nuclei were stained with Hoechst solution. Then, the cell count and the fluorescence were analyzed with Cytation^TM^. Four pictures containing 20–30 cells per field of view were taken for each well, and each condition was represented by at least three internal replicates. Per experiment, three independent culture plates were analyzed. In negative control wells, the addition of primary antibody was omitted. Micrographs were taken with Cytation^TM^. Cell counting and fluorescence analysis were performed with ImageJ software 1.5 [64]. The obtained relative fluorescent units (RFU) per field were divided by the number of cells. In some experiments, the fluorescence of exogenously added CXCL4(L1) was normalized against the baseline fluorescence.

### 4.6. Quantitative Real-Time PCR (qPCR)

VSMCs were seeded in triplicate in 6-well plates at a density of 120,000 cells/well in DMEM with 20% FCS. The next day cells were treated without or with CXCL4 or CXCL4L1 at (1 µg/mL) in DMEM, supplemented with 2.5% FCS and kept in the incubator for 24 or 72 h. After that, VSMCs were lysed in TRI reagent and total RNA extracted, quantified at 260 nm and reverse transcribed using iScript for RT-qPCR (Appendix A). Gene expression levels were analyzed by real-time quantitative PCR (qPCR) on a LightCycler 480 (Roche Applied Science, Basel, Switzerland). Amplification reactions were carried out in a volume of 10 μL including 45 ng of total cDNA, 5.5 μL of SYBR-dTTP mix and 62.5 nM of each primer. Amplification specificity was confirmed by melting curves and curves were analyzed with LinRegPCR software (11.0) [65]. Relative quantification of expression was achieved by plotting Cq ratios.

### 4.7. Proliferation Assay Using the xCELLigence System

Cells were seeded into 96-well electronic microtiter plates (E-plate^®^) at 3000 cells/well in DMEM and observed for 9 days. Cell growth (impedance value) was plotted as a unitless parameter (Cell Index) over time. The rate of cell growth was determined by calculating the slope of the curve between the end of the lag phase (50 h) and the start of the stationary confluent phase (100 h).

### 4.8. Reactive Oxygen Species (ROS) Assay

VSMCs were cultured in DMEM, supplemented with 2.5% FCS overnight, incubated with the DCFDA dye for 30 min at 37 °C and stimulated with the vehicle and stimuli (see above). ROS production was measured for 6 h by Cytation^TM^ under at 5% CO_2_ and 37 °C. The data were analyzed by calculating area under the curve (AUC) and normalized to the cell count.

### 4.9. Calcification Assay

VSMCs were seeded in the 48-well plates (10,000 cells/well), cultured in DMEM/20% FCS and treated with or without increased calcium (2.7 mM) and phosphate (2.5 mM) or calcium/phosphate in combination with CXCL4 or CXCL4L1 (1 μg/mL or 10 μg/mL). To accelerate calcification, serum was reduced to 0.5% FCS in all conditions and incubated for 24 h. Calcification was quantified using a commercial o-cresolphthalein assay and normalized to protein content.

### 4.10. Statistical Analysis

Statistical analysis was performed using Graphpad Prism 9.0.0 (Graphpad Software, San Diego, CA, USA). Data are presented as means ± SD and were compared by (non-)parametric 1-way or 2-way ANOVA. Appropriate correction for multiple comparisons was achieved depending on the statistical test. Differences with *p* < 0.05 were considered as statistically significant. Each experiment was independently repeated at least three times, as indicated for each experiment in the figure legends.

## Figures and Tables

**Figure 1 ijms-23-00580-f001:**
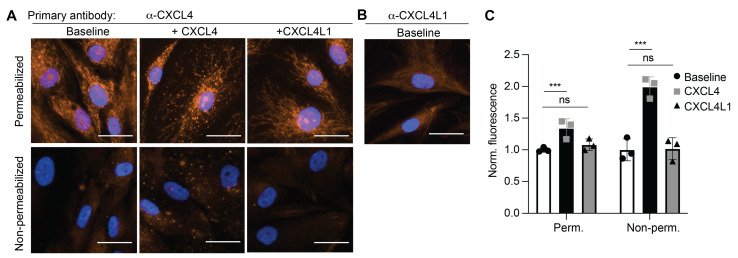
Uptake of platelet factor 4 (CXCL4) into vascular smooth muscle cells (VSMCs). CXCL4, CXCL4L1 (at 0.5 µg/mL) or buffer (baseline) was added to VSMC for 1 h at 37 °C, washed with heparin and stained for CXCL4 (**A**) or CXCL4L1 (**B**) with or without prior permeabilization. Representative micrographs showing CXCL4 (**A**) or CXCL4L1 (**B**) staining (orange) and nuclei (blue). Scale bar: 50 µm. (**C**) Quantification of fluorescence expressed normalized to baseline fluorescence, of 4 images per well (3 wells per experiment of 3 separate experiments) divided by cell count, measured using Cytation™. Non-significant: ns, *** *p* < 0.001, *n* = 3, two-way ANOVA with Dunnett’s post-test.

**Figure 2 ijms-23-00580-f002:**
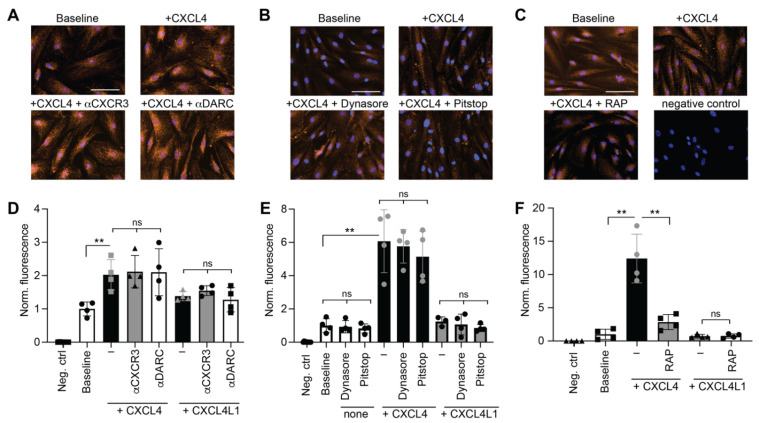
Molecular determinants of CXCL4 uptake. CXCL4, CXCL4L1 (at 0.5 µg/mL) or buffer (baseline) was added to VSMCs in the presence of indicated compounds for 1 h at 37 °C, washed with heparin and stained for CXCL4 with prior permeabilization. (**A**,**D**) Representative images and quantitation of chemokines, co-incubated with antibodies against CXCR3 or DARC and resulting intracellular CXCL4 was expressed normalized to baseline fluorescence, measured using Cytation™. (**B**,**E**) Representative images and quantitation of chemokines, co-incubated with Dynasore (80 µM) or PitStop2 (15 µM) and resulting intracellular CXCL4 was expressed normalized to baseline fluorescence, measured using Cytation™. (**C**,**F**) Representative images and quantitation of chemokines, co-incubated with RAP (200 nM) and resulting intracellular CXCL4 was expressed normalized to baseline fluorescence, measured using ImageJ. Scale bar: 100 µm. Non-significant: ns, ** *p* < 0.01, *n* = 4, one-way analysis of variance (ANOVA) with Tukey’s post-test (**D**,**E**) or with Dunnett’s post-test (**F**). Representative figures of CXCL4L1 treatment are shown in Appendix A.

**Figure 3 ijms-23-00580-f003:**
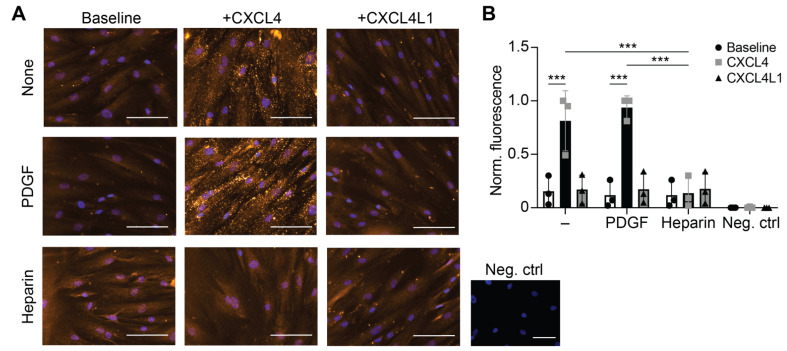
Influence of VSMC phenotype on chemokine uptake. Prior to the addition of CXCL4, CXCL4L1 (at 0.5 µg/mL) or buffer, VSMC were cultured in the presence of PDGF-BB (20 ng/mL, 3 days) or heparin (200 U/mL, 5 days). After incubation with chemokines for 1 h at 37 °C, cells were washed with heparin and stained for CXCL4 with permeabilization. The negative control is without anti-CXCL4 antibody. (**A**) Representative micrographs showing CXCL4 staining. Scale bar: 100 μm. (**B**) Quantification of fluorescence normalized by minimum and maximum, four images per well (three wells per experiment of three separate experiments). *** *p* < 0.001, *n* = 3, two-way ANOVA with Dunnett’s post-test.

**Figure 4 ijms-23-00580-f004:**
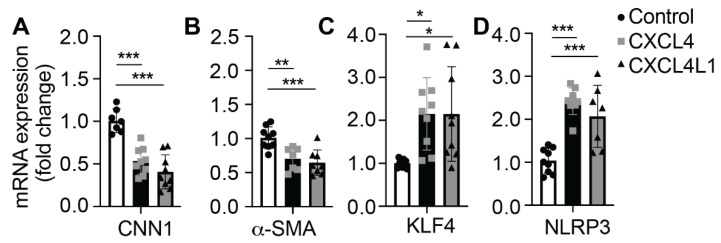
Expression of mRNA by quantitative real-time PCR. Primary VSMCs were cultured at 120,000 cells/well in Dulbecco’s modified Eagle medium (DMEM) with 20% FBS prior to treatment with or without CXCL4 or CXCL4L1 at 1 µg/mL in DMEM with 2.5% FBS for 24 h. Relative gene expression was expressed as fold change of (**A**) calponin (CNN1), (**B**) α-smooth muscle actin (α-SMA), (**C**) KLF4, (**D**) NLRP3. * *p* < 0.05, ** *p* < 0.01, *** *p* < 0.001, *n* = 7–9 independent experiments, one-way ANOVA with Dunnett’s post-test.

**Figure 5 ijms-23-00580-f005:**
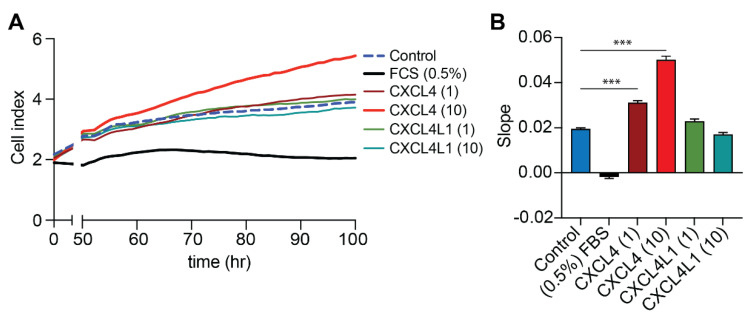
Influence of CXCL4 and CXCL4L1 on proliferation of VSMCs. Cell growth was monitored in 96-well electronic microtiter plates for 9 days and (**A**) plotted as mean cell index over time (*n* = 5). (**B**) The rate of cell growth expressed as slopes between 50 and 100 h of proliferation from the curves from (**A**), mean ± SD. *** *p* < 0.001, *n* = 5, one-way ANOVA with Dunnett’s post-test.

**Figure 6 ijms-23-00580-f006:**
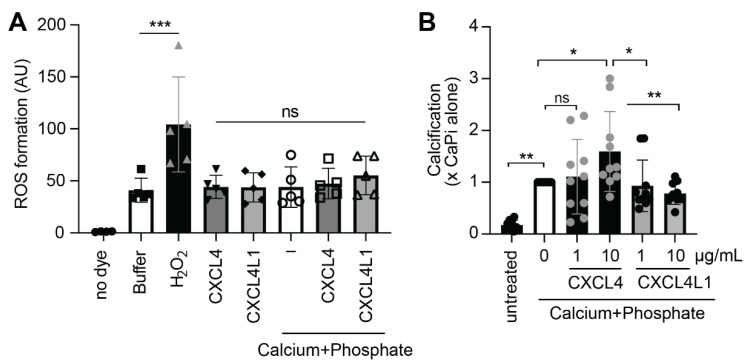
Formation of reactive oxygen species (ROS) and calcium mineral crystals in VSMCs. (**A**) Cultured VSMCs in DMEM/2.5% FCS overnight, loaded with or without DCFDA dye and stimulated with vehicle, CXCL4 or CXCL4L1 (10 μg/mL) in the presence or absence of calcium (2.7 mM) and phosphate (2.5 mM). ROS formation was expressed as arbitrary fluorescence units (AU). Non-significant: ns, *** *p* < 0.001, *n* = 4, one-way ANOVA with Sidak post test. (**B**) Cultured VSMCs in DMEM/20% FCS treated with or without calcium (2.7 mM) and phosphate (2.5 mM) in the presence of vehicle, CXCL4 or CXCL4L1 (1 and 10 μg/mL). Calcification was expressed as fold control in the absence of chemokines (white bar). * *p* < 0.05, ** *p* < 0.01, *** *p* < 0.001, *n* = 10 independent experiments, one-way ANOVA with Sidak post test.

**Figure 7 ijms-23-00580-f007:**
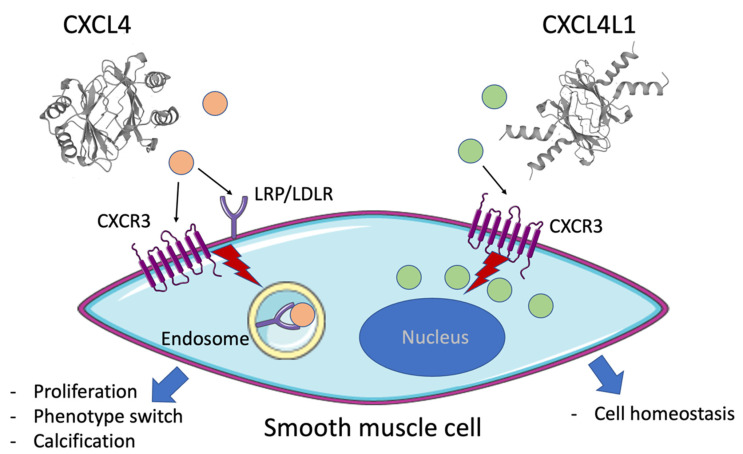
Schematic summary of the proposed action of CXCL4 and CXCL4L1 on smooth muscle cells. CXCL4 (orange dot and ribbon structure pdb: 1PF9Q) is taken into the cells by the action of the low-density lipoprotein (LDL) receptor family proteins or lipoprotein receptor-related protein (LRP) into endosomes. CXCL4-induced signals might be transmitted through both CXCR3 [4] and LRP [20], leading to proliferation, phenotypic alterations and calcification. CXCL4L1 (green dot and ribbon structure pdb: 4HSV) is present endogenously and is not taken up by SMC, but may induce cell signaling through CXCR3 [6]. It is postulated that CXCL4L1 might be involved in cell homeostatic processes.

## Data Availability

Data are available upon reasonable request from the corresponding author. Reagents and detailed methods of all procedures are provided in the Appendix A.

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
