# Peer review of "Differential Effects of Platelet Factor 4 (CXCL4) and Its Non-Allelic Variant (CXCL4L1) on Cultured Human Vascular Smooth Muscle Cells"

_ijms, 2022, doi:10.3390/ijms23020580_

Round 1

Reviewer 1 Report

The authors nicely showed that CXCL4 is taken from VSMCs up whereas CXCL4L is already found in VSMCs. In addition, only CXCL4 can induce proliferation and calcification of VSMCs. In contrast, there where no differences found in the expression of various inflammatory cytokines and transcription factors and the production of ROS. The data are clearly presented, but I have some revisions that are necessary before the manuscript can be published.

  1. In Figure 1A, for me it's unclear whether "control" means buffer-treated cells, as in the left image of the upper panel a clear staining of CXCL4 is visible, which is not represented in the quantification in 1C. In addition, it's not described how many cells were counted for the quantification of the ratio. This should be added in all these kind of figures.
  2. In Figure 2, the authors incubate the chemokines with the blocking antibodies against the receptors. In my opinion, this doesn't make any sense. This experiment has to be repeated with an incubation of the cells with the blocking antibodies for 1 hour and then incubation with the chemokines and subsequently analysis of the uptake as performed before.
  3. In Figure 4, a clear reduction of IL-6 mRNA is shown whereas on protein level this reduction is not measurable. The authors should comment on this at least in the discussion.
  4. In the discussion, the authors claim distinct roles, but in my opinion in the manuscript only two points show differences. This is first the uptake, where it is not clear whether this has any effect onthe VSMCs, and this is second the calcification. But it's not that CXCL4 has a more pro-inflammatory role which CXCL4L has not! For this statement further experiments are necessary.
  5. In the rest of the discussion the authors state that various results a different from their own previous studies and from studies from others and this could be the result of using VSMCs from just one patient. Therefore, at least some of the experiments have to be repeated with VSMCs from an other patient to really be sure that the results which they describe are comparable. Then the statement that CXCL4 and CXCL4L 

Author Response

The authors nicely showed that CXCL4 is taken from VSMCs up whereas CXCL4L is already found in VSMCs. In addition, only CXCL4 can induce proliferation and calcification of VSMCs. In contrast, there where no differences found in the expression of various inflammatory cytokines and transcription factors and the production of ROS. The data are clearly presented, but I have some revisions that are necessary before the manuscript can be published.

We thank the reviewer for the encouraging comments and for the thorough assessment of our study. We are happy to address the comments below and we believe that the study has improved in the process.

In Figure 1A, for me it's unclear whether "control" means buffer-treated cells, as in the left image of the upper panel a clear staining of CXCL4 is visible, which is not represented in the quantification in 1C. In addition, it's not described how many cells were counted for the quantification of the ratio. This should be added in all these kind of figures.

We realize that the presentation of this figure could have been clearer. In this experiment “control” equals buffer-treated cells, ie. no exogenously added CXCL4(L1). The presence of CXCL4 is indeed clearly visible and was quantified in figure 1C (white bar) which is comparable to the fluorescence intensity of CXCL4L1-treated cells (grey bar). We have renamed the “control” conditions to “baseline”.

In immunocytochemical experiments usually 4 pictures were taken for each well and each condition was represented by at least 3 internal replicates. In total 3 culture plates were analysed. On average there were around 20-30 cells/field of view (depending on the picture). For each well, the RFU was normalized to the cell count from each separate pictures and then averaged (the exact cell count is different for each well). We have adjusted figure 1 to make it clearer and added the information in its legend and in the methods section, page 11, section 4.5.

In Figure 2, the authors incubate the chemokines with the blocking antibodies against the receptors. In my opinion, this doesn't make any sense. This experiment has to be repeated with an incubation of the cells with the blocking antibodies for 1 hour and then incubation with the chemokines and subsequently analysis of the uptake as performed before.

We thank the reviewer for this concern and apologize for omitting this detail in the methods section: cells were pre-treated with 10 µg/ml CXCR3 or DARC blocking antibody (or other inhibitors) for up to 30 minutes. Then, the fresh medium (DMEM + 0.5% FCS) containing 10 µg/ml of blocking antibodies (anti-CXCR3 or anti-DARC) together with 500 ng/ml of the chemokines (CXCL4 or CXCL4L1) was added to the cells and the culture plates were incubated for additional 1 hour at 37°C. Given the high affinity of the antibodies used, we believe that 30 minutes preincubation with antibodies and then 1 hour co-incubation was sufficient to block the interactions between chemokines and their receptors. We have clarified this now in the methods section on page 10, line 378.

In Figure 4, a clear reduction of IL-6 mRNA is shown whereas on protein level this reduction is not measurable. The authors should comment on this at least in the discussion.

The mRNA expression of IL-6 was indeed reduced after the treatment with both chemokines. Due to the high sensitivity of the qPCR technique, it is possible that we were able to see subtle differences in mRNA levels, which were still undetectable at the protein expression level. We did measure the secretion profile of IL-6 by VSMCs after the treatment with the chemokines and observed that there was no increase in cytokine secretion. Although this is somewhat counterintuitive, protein expression is regulated on several levels, and we have witnessed discordances in mRNA and protein expression more often.

For the data concerning IL-1ß and IL-6, we realize that these data are contradictory and not in line with previous findings in the literature (e.g Shi et al., doi:10.1182/blood-2012-09-454710). Since these data are not obligatory to support our conclusion, we have decided to remove them from the paper.

In the discussion, the authors claim distinct roles, but in my opinion in the manuscript only two points show differences. This is first the uptake, where it is not clear whether this has any effect on the VSMCs, and this is second the calcification. But it's not that CXCL4 has a more pro-inflammatory role which CXCL4L has not! For this statement further experiments are necessary.

We agree that such statement requires more experimental work. In the discussion we were trying to point into a direction in which we believe the CXCL4 biology is going, taking into account other previous relevant scientific reports. At present our results do not clearly demonstrate that CXCL4 has a stronger pro-inflammatory role than CXCL4L1. We do postulate that CXCL4L1, due to its endogenous expression, might exert a role in cell homeostasis. We have now adjusted the section in the discussion in the revised manuscript on page 9.

In the rest of the discussion the authors state that various results a different from their own previous studies and from studies from others and this could be the result of using VSMCs from just one patient. Therefore, at least some of the experiments have to be repeated with VSMCs from an other patient to really be sure that the results which they describe are comparable. Then the statement that CXCL4 and CXCL4L 

This is a keen observation by the reviewer. We clearly stated in the methods sections that the SMCs used in this study were isolated from a single patient. This might be regarded as a limitation in our manuscript. On the other hand, we consider using early passage primary cells isolated under well controlled conditions in our laboratory as a strength. In addition, the donor -though not known by name for ethical reasons- was well defined. This is in contrast with SMCs obtained from commercial sources, of which donor characteristics and culture conditions are poorly disclosed, if at all. Finally, many well established studies are done with cell lines, of which most are derived from a single (cancer) patient or donor.

Reviewer 2 Report

The study „Differential effects of platelet factor 4 (CXCL4) and its non-allelic variant (CXCL4L1) on cultured human vascular smooth muscle cells “ by Dawid M. Kaczor et al investigated the effects of CXCL4 and CXCL4L1 on human VSMCs. The authors focused on the possible uptake of these chemokines and the mechanism which regulates this process. Moreover, expression of some contractile, inflammatory and transcriptional factors was studied by qPCR. Finally, influence of CXCL4 and CXCL4L1 on proliferation as well as on calcification was probed.

The present article reveals many open questions. Main concerns are listed subsequently:

  • Figure 1, legend line 100-104: Labelling in the figure legend is wrong. Incubation with CXCL4L1 is indicated as (B). However, (B) is showing baseline level of CXCL4L1 without stimulation.
  • Figure 1: Why the authors decided for a concentration of 0,5 µg/ml? Is this a physiological concentration? Please try other concentrations also to see a dose-dependent effect.
  • Figure 1: The data are not fully convincing since it is only n=3 and moreover only one read out. E.g. ELISA of the remaining medium should be performed for CXCL4 to detect lower levels in the medium where higher uptake by VSMC was observed.
  • Figure 2: Please provide representative micrographs for these data also (at least in the supplement).
  • Figure 2A: Isn`t it the same method and set-up like in Figure 1? Why are the scales then so different? E.g. Figure 1 shows values above 0.5 RFU/cell count where as Figure 2A only shows less than 0.15 RFU/cell count. We should expect same extent of CXCL4 uptake at least in the control groups. Please explain.
  • Figure 2A: Results of samples which were co-incubated with antibodies against CXCR3 or DARC but not incubated with CXCL4 or CXCL4L1 should be included (compare Figure 2B “none”-group).
  • Figure 2B: Different scale is used here (RFU x 10^-2) in comparison to Figure 2A. Please use the same scale everywhere. Why do the authors not show a negative control here?
  • Figure 2C: Why again changing the measurement here? Now the authors show the integrated density (10^7) / cell count. The experimental set-up was not changed. Why the authors not showing RFU/cell count then also here?
  • Figure 2C: Again “none”-group is missing which reflects a sample treated with RAP but without any chemokines.
  • Figure 3A: Again, experimental set-up is the same compared to Figure 1 and 2, isn´t it? Why the authors cannot provide the same read-out like in Figure 1 (colored micrographs, scale 50 µM instead of 100 µM)?
  • Figure 3B: Now another scale is used again. Please decide for one scale / read-out (e.g. RFU/cell count).
  • Figure 3: The conditions of pre-treatment with PDGF and heparin differs a lot (3 days vs 5 days+overnight starvation). Were the VSMC in the control group of the same age? Was an overnight starvation also performed for PDGF and the control group? If not, one control group for both conditions is not sufficient here. For every condition a suitable control group should be included.
  • Figure 4: mRNA data are not convincing for several reasons:
    • Why CXCL4 and CXCL4L1 concentration is now increased to 1 µg/ml? Before 0,5 µg/ml was always used. In general, different concentrations should be checked  here to see a dose-dependent effect. Since significant uptake of CXCL4 was observed at a concentration of 0.5 µg/ml, this concentration should be at least also used here.
    • Why the authors decided for n=7-9 here? Since there is a huge standard deviation for KLF4 and NLRP3 the authors should explain this.
    • mRNA data alone are not convincing. For IL-1ß and IL-6 ELISA data should be included (for IL-6 ELISA data are provided, however they show contradictory results to mRNA data). For CNN1 and a-SMA Western Blot data are needed.
    • To show mRNA-upregulation of transcription factors KLF4 and NLRP3 alone is for sure not sufficient. Here, studying downstream targets is heavily needed. E.g.  upregulation of KLF4 is considered to be connected with less VSMC proliferation which would be contradictory to Figure 5.
    • When studying these aims this should be sustained by knockdown approaches of CXCL4L1, e.g. via siRNA approaches. Moreover, co-treatment with CXCL4 and RAP should be included here.
  • Figure 5: How was the slope measured? As a mean value over the whole time or at a specific time point? From looking at the curves difference between Control and CXCL4 (1) curve is not convincing. Why the authors picked 100 hrs at the last time point? Until 70 hrs even less proliferation is observed in this group. 10 µg/ml is a very high concentration in comparison to concentration which were used in all the other experiments. A concentration inbetween should be used to sustain the data. Knockdown approach should be included.
  • Figure 6: Experiments should be repeated by including a new group with RAP to prevent the VSMCs from taking up CXCL4.
  • Figure 6: In line 347-349 the authors state that “addition of CXCL4L1 showed a non-significant trend towards a reduction”. First, by looking at Figure 6B we cannot see a reduction but an enhancement of calcification in comparison to the control group. How the authors explain this effect? Second, in the figure the authors show a significant reduction (in the discussion they say “non-significant”) when they compare CXCL4 group to CXCL4L1 group. However, the statement that addition of CXCL4L1 leads to a reduction of calcification is misleading. For this statement the authors need a co-treatment of both, CXCL4 and CXCL4L1 in one group. Here they can only conclude, that CXCL4L1 also leads to augmented calcification but less than CXCL4 group.
  • A model of CXCL4 and CXCL4L1 signalling in VSMC should be included.
  • The authors discuss CXCL4L1 as a potential “physiologic regulatory mechanism to maintain vascular homeostasis”. This should be investigated, e.g. with a knockdown approach of CXCL4L1 and subsequent investigation of VSMC and their potential alteration in secretion of chemokines important for endothelial cell proliferation and angiogenesis. These data should be included in the study.
  • Reagents: The authors should provide evidence that they really isolated / purified CXCL4 / CXCL4L1 (e.g. via ELISA). Did they check for endotoxins in these reagents?

Author Response

The study „Differential effects of platelet factor 4 (CXCL4) and its non-allelic variant (CXCL4L1) on cultured human vascular smooth muscle cells“ by Dawid M. Kaczor et al investigated the effects of CXCL4 and CXCL4L1 on human VSMCs. The authors focused on the possible uptake of these chemokines and the mechanism which regulates this process. Moreover, expression of some contractile, inflammatory and transcriptional factors was studied by qPCR. Finally, influence of CXCL4 and CXCL4L1 on proliferation as well as on calcification was probed.

We thank the reviewer for the thorough and extensive assessment of our work. We have done our best to address the constructive criticisms and questions within the limited time for revisions granted to us by the journal and we are confident that our manuscript improved considerably in the process.

The present article reveals many open questions. Main concerns are listed subsequently:

Figure 1, legend line 100-104: Labelling in the figure legend is wrong. Incubation with CXCL4L1 is indicated as (B). However, (B) is showing baseline level of CXCL4L1 without stimulation.

We realize that the presentation of this figure could have been clearer. In this experiment the staining in Figure 1A was performed using an antibody against both CXCL4 and CXCL4L1 and in Figure 1B the staining was performed with a specific antibody against CXCL4L1. The indications above the panels showed exogenously added chemokines or vehicle “control”, which equals buffer-treated cells, ie. no exogenously added CXCL4(L1). We have updated the figure and the legend for more clarity in the revised version of this study.

Figure 1: Why the authors decided for a concentration of 0,5 µg/ml? Is this a physiological concentration? Please try other concentrations also to see a dose-dependent effect.

Previous work performed in our group proved 0.5 µg/ml of CXCL4 and CCL5 to be optimal for internalization experiments in endothelial cells (Dickhout et al. doi: 10.3390/ijms22147332) and also yielded robust results in leukocyte recruitment experiments, see also: von Hundelshausen et al., Blood 2005, Koenen et al., Nat. Med. 2009, von Hundelshausen et al., Sci. Trans. Med. 2017). In another paper, the gene expression, IL-6 secretion, proliferation and migration was measured after treatment with 0.1 – 1 µg/ml (doi: 10.1182/blood-2012-09-454710). We have tested several doses (also 1 and 10µg/mL, see figure 4, 5, and 6) in this study, and show dose-dependent effects of CXCL4. Since platelets contain large amounts of CXCL4 and CXCL4L1, local concentrations can very well exceed 10 µg/mL at sites of platelet aggregation (e.g. Brandt et al., doi: 10.1034/j.1600-065x.2000.17705.x.). In our own recent study, CXCL4 concentrations of approx. 5µg/mL (600nM) were readily reached in suspensions of activated platelets with physiologic platelet counts (Dickhout et al., doi: 10.1371/journal.pone.0244736) and plasma concentrations of mice with carotid artery injury showed plasma concentrations of up to 1.4 µg/mL (Shi et al., doi:10.1182/blood-2012-09-454710), indicative of higher concentrations at the site of injury.

Figure 1: The data are not fully convincing since it is only n=3 and moreover only one read out. E.g. ELISA of the remaining medium should be performed for CXCL4 to detect lower levels in the medium where higher uptake by VSMC was observed.

Actually, the image data are the result of many quantified cells. In immunocytochemical experiments usually 4 pictures were taken for each well and each condition was represented by at least 3 internal replicates. In total 3 culture plates were analysed. On average there were around 20-30 cells/field of view (depending on the picture). For each well, the RFU was normalized to the cell count from each separate pictures and then averaged. We have now indicated this in the figure legends and in the method section on page 11, section 4.5.

Figure 2: Please provide representative micrographs for these data also (at least in the supplement).

We have now added representative pictures as requested in Figure 2 and in the supplementary figures S2 and S3. In figure 2, we chose to only include the images of CXCL4 uptake to avoid overcrowding the figure. The CXCL4L1 images were added as Figure S3.

Figure 2A: Isn`t it the same method and set-up like in Figure 1? Why are the scales then so different? E.g. Figure 1 shows values above 0.5 RFU/cell count where as Figure 2A only shows less than 0.15 RFU/cell count. We should expect same extent of CXCL4 uptake at least in the control groups. Please explain.

This is well noted by the reviewer. The different values between the panels are simply due to day-to-day experimental variation, which arises by the density, counts and passage number of the cells, slight alterations in microscope settings for optimal illumination and detection, and variations in primary and secondary antibody binding. We have chosen to perform independent assays on different days. Since the trend of our data consistently goes in the same direction (i.e. significant uptake of CXCL4) in each independent experiment, we feel that the results of the experiments are valid. We have now normalized all scales of the CXCL4 uptake data in the main manuscript to the baseline CXCL4L1 fluorescence (without any exogenous chemokine, or treatment). This indeed allows for better comparison between the figures.

Figure 2A: Results of samples which were co-incubated with antibodies against CXCR3 or DARC but not incubated with CXCL4 or CXCL4L1 should be included (compare Figure 2B “none”-group).

With all due respect, this experiment would only result in the visualization of the baseline CXCL4L1 expression in the cells (cf. Figure 1C and Figure 2A,B white bars), since blocking these receptors would not affect basal CXCL4L1 content. Thus, including such data would not have any added value to the story. Neither as data, nor as control.

Figure 2B: Different scale is used here (RFU x 10^-2) in comparison to Figure 2A. Please use the same scale everywhere. Why do the authors not show a negative control here?

Figure 2C: Why again changing the measurement here? Now the authors show the integrated density (10^7) / cell count. The experimental set-up was not changed. Why the authors not showing RFU/cell count then also here?

We thank the reviewer for pointing this out and we have now made the axis-scales of all figures in the main text consistent by normalizing the fluorescence values with the values of the baseline CXCL4L1 fluorescence (without any exogenous chemokine, or treatment). We have also added the requested negative control data to Figure 2B.

Figure 2C: Again “none”-group is missing which reflects a sample treated with RAP but without any chemokines.

With all due respect, this experiment would only result in the visualization of the baseline CXCL4L1 expression in the cells and would not have any added value to the story. Neither as data, nor as control.

Figure 3A: Again, experimental set-up is the same compared to Figure 1 and 2, isn´t it? Why the authors cannot provide the same read-out like in Figure 1 (colored micrographs, scale 50 µM instead of 100 µM)? Figure 3B: Now another scale is used again. Please decide for one scale / read-out (e.g. RFU/cell count).

We thank the reviewer for pointing this out. These experiments were performed using a lower magnification power as those presented in figure 1, which does not influence the experimental outcome. Following the reviewer’s suggestion, we have now exchanged all black and white pictures with color pictures and also adjusted the scale of Figure 3B to match (revised) Figures 1B and 2.

Figure 3: The conditions of pre-treatment with PDGF and heparin differs a lot (3 days vs 5 days+overnight starvation). Were the VSMC in the control group of the same age? Was an overnight starvation also performed for PDGF and the control group? If not, one control group for both conditions is not sufficient here. For every condition a suitable control group should be included.

The treatment regimes were adapted from our previous studies (e.g. Furmanik et al., doi: 10.1161/CIRCRESAHA.119.316159.). All cells on one plate were of the same age. Overnight starvation was performed for every condition.

Protocol: Human VSMC were seeded in a 96-well plate at a density of 3.000 cells/well and incubated in DMEM, supplemented with 20% FCS overnight. Then, a selection of wells was pre-incubated without or with heparin (200 U/ml, 5 days), the rest of the cells were left untreated for 2 days. After these 2 days, PDGF (20 ng/ml, 3 days) was added to some wells. After 5 days, all cells were starved at the same time in DMEM with 0.5% FCS overnight. The next day CXCL4 or CXCL4L1 was added at a final concentration of 500 ng/ml (in DMEM with 0.5% FCS) and incubated for 1 hour. The amount of CXCL4 chosen was sufficient to obtain a robust staining.

Figure 4: mRNA data are not convincing for several reasons:

Why CXCL4 and CXCL4L1 concentration is now increased to 1 µg/ml? Before 0,5 µg/ml was always used. In general, different concentrations should be checked here to see a dose-dependent effect. Since significant uptake of CXCL4 was observed at a concentration of 0.5 µg/ml, this concentration should be at least also used here.

We followed the procedure that was published in a previous paper in which the gene expression was measured after treatment with 1 µg/ml of CXCL4 (DOI: 10.1182/blood-2012-09-454710) and we reasoned that this would be feasible to obtain measurable effects. We decided to use only one concentration of the chemokines for various reasons: 1. The very large amount of slowly growing primary cells required to extract sufficient RNA for qPCR experiments. 2. This amount is multiplied by two time points (24h and 72h), 3. and by the control and two chemokines used, 4. each experiment performed with 3 internal replicates and finally 5. we decided to examine 6 molecular markers.

Why the authors decided for n=7-9 here? Since there is a huge standard deviation for KLF4 and NLRP3 the authors should explain this.

In our qPCR studies, 3 plates with 3 internal replicates were analysed and we decided to show the datapoint of each replicate, resulting in 9 points in the graph. However, some replicates showed abnormalities (e.g. in the melting curve) and had to be excluded, which resulted in n= 7 in some instances.

Due to the exponential nature of PCR, minor pipetting errors are also amplified, and may inherently result in variation after the calculations, thus explaining the occurrence standard deviations. On the other hand, the expression of these markers might vary more than the other markers between the independent experiments. Although we agree that the standard deviations of the KLF4 and NLRP3 transcripts are higher than those seen with the other markers, we do think that these are not abnormally high for qPCR data and the resulting expression levels after CXCL4(L1) treatment are still well separated from the control conditions. We have briefly referred to this on page 5, line 170.

mRNA data alone are not convincing. For IL-1ß and IL-6 ELISA data should be included (for IL-6 ELISA data are provided, however they show contradictory results to mRNA data). For CNN1 and a-SMA Western Blot data are needed.

For the data concerning the inflammatory markers IL-1ß and IL-6, we agree with the reviewer that these data are contradictory and not in line with previous findings in the literature (e.g Shi et al., doi:10.1182/blood-2012-09-454710). Since these data are not obligatory to support our conclusion, we have decided to remove them from the paper.

For the calponin and SMA data we followed the previously published experimental setup and we were able to confirm these data (doi:10.1182/blood-2012-09-454710). Thus, we hope that the reviewer agrees that further experimental elaboration would not significantly add to the already presented findings.

To show mRNA-upregulation of transcription factors KLF4 and NLRP3 alone is for sure not sufficient. Here, studying downstream targets is heavily needed. E.g.  upregulation of KLF4 is considered to be connected with less VSMC proliferation which would be contradictory to Figure 5.

By presenting a modulation of KLF4 and NLRP3 levels by CXCL4 and CXCL4L1, we think that we provided some new insight in the biology of these chemokines, showing for the first time an effect of CXCL4 on NLRP3 expression. In addition, the data on the effects of CXCL4 on KLF4 expression are in line with a previous study by Shi et al., (DOI: 10.1182/blood-2012-09-454710), in which CXCL4 was found to stimulate the proliferation of VSMCs and increased KLF4 expression. We think that our findings concerning proliferation and KLF4 expression are not contradicting.

When studying these aims this should be sustained by knockdown approaches of CXCL4L1, e.g. via siRNA approaches. Moreover, co-treatment with CXCL4 and RAP should be included here.

Approaches using siRNA would be ideal, however we wanted to keep this experiment simple in order to screen a small panel of the many markers that might be affected by CXCL4/CXCL4L1. In addition, siRNA transfections are complicated in primary cells such as SMC and results might be confounded by phenotypic and expression changes induced by liposome- or viral-mediated transfection methods. For the aspect of CXCL4 activity on SMC, the actions of RAP have been published before. For CXCL4L1, additional studies are needed to pinpoint the receptor and molecular pathway responsible for the effects that we have observed. We have discussed this now on page 9, line 337.

Figure 5: How was the slope measured? As a mean value over the whole time or at a specific time point? From looking at the curves difference between Control and CXCL4 (1) curve is not convincing. Why the authors picked 100 hrs at the last time point? Until 70 hrs even less proliferation is observed in this group. 10 µg/ml is a very high concentration in comparison to concentration which were used in all the other experiments. A concentration inbetween should be used to sustain the data. Knockdown approach should be included.

The experiment was done in an automated (Xcelligence®) system that enabled continuous monitoring of cell growth based on impedance. The curves shown in Figure 5A are means (n=5) of the obtained growth profiles and were shown between the lag phase (0–50 hours) and the stationary confluent phase (after 100 hours). The slope was calculated as a mean value over this timespan (50–100 hours). The mean slope of the CXCL4 (1µg/ml) group clearly is higher than that of the control group, the CXCL4 curve just starts at a slightly lower cell index.

The demanding VSMCs were incubated on the gold microplates for 9 days with low serum and the medium was not replaced, so this limited the duration of the experiment, reaching a stationary phase after 100 hours. We decided that 2 concentrations of the two chemokines tested would by sufficient and we chose the 2 concentrations that we used for all functional studies throughout this paper: 1 µg/ml and 10 µg/ml (we used 0.5 µg/ml only for the internalization experiments).

The quantification of the growth shows a significant mitogenic effect already at 1 µg/ml of CXCL4, which is dose-dependent and stronger at 10 µg/ml. We chose this high concentration (but not exceptionally high, see previous reply) to demonstrate that even at that concentration CXCL4L1 did not have an effect. For these reasons, we do not see the necessity to add an intermediate concentration (e.g. 5µg/ml).

We have adjusted figure 5A to have the axis start with zero to better indicate that the slopes have not been calculated from the start and we have also indicated this more clearly in the methods section on page 11, line 426 and in the legend of the figure.

Figure 6: Experiments should be repeated by including a new group with RAP to prevent the VSMCs from taking up CXCL4.

As mentioned above, functional studies of CXCL4 and RAP have already been published (Shi et al., doi:10.1182/blood-2012-09-454710). We have added novel findings in a sense that RAP also prevents the uptake of CXCL4, something that has not been shown before.

Figure 6: In line 347-349 the authors state that “addition of CXCL4L1 showed a non-significant trend towards a reduction”. First, by looking at Figure 6B we cannot see a reduction but an enhancement of calcification in comparison to the control group. How the authors explain this effect? Second, in the figure the authors show a significant reduction (in the discussion they say “non-significant”) when they compare CXCL4 group to CXCL4L1 group. However, the statement that addition of CXCL4L1 leads to a reduction of calcification is misleading. For this statement the authors need a co-treatment of both, CXCL4 and CXCL4L1 in one group. Here they can only conclude, that CXCL4L1 also leads to augmented calcification but less than CXCL4 group.

We apologize for the unclarity, and we realize that the presentation of the data was misleading. We have now updated Figure 6 and added a bar representing 0 µg/ml chemokine and renamed the “control” bar as untreated. Here, the “untreated” group means that the cells were not treated with Ca/Pi, so the calcification is zero. The 0 µg/ml the graph represents the level of calcification that was present after the treatment with Ca/Pi but no chemokines. All the values are displayed as a fold change compared to Ca/Pi only treated cells. We have also removed the sentence that CXCL4L1 and CXCL4 might act in a complementary fashion, of which we realise that it is not fully supported by the data.

A model of CXCL4 and CXCL4L1 signalling in VSMC should be included.

We have now added a scheme summarizing what is currently known in literature and the current findings of this study as Figure 7 and a brief discussion on page 9, line 343.

The authors discuss CXCL4L1 as a potential “physiologic regulatory mechanism to maintain vascular homeostasis”. This should be investigated, e.g. with a knockdown approach of CXCL4L1 and subsequent investigation of VSMC and their potential alteration in secretion of chemokines important for endothelial cell proliferation and angiogenesis. These data should be included in the study.

We thank the reviewer for this suggestion. Indeed, CXCL4L1 knockdown would be an elegant tool to investigate its role in maintaining vascular homeostasis. In addition, (siRNA) transfections are complicated in primary cells such as SMC and might be confounded by phenotypic and expression changes induced by liposome- or viral-mediated transfection methods. We have chosen to discuss this experimental option, perhaps as modified iPSC, for future studies in the discussion on page 9, line 337.

Reagents: The authors should provide evidence that they really isolated / purified CXCL4 / CXCL4L1 (e.g. via ELISA). Did they check for endotoxins in these reagents?

The isolation of CXCL4 from expired platelet packs and the expression and isolation of CXCL4L1 from E. coli is a routine procedure in our laboratory. We have now added chromatograms and mass spectra of the final products as proof of purity and protein identity in supplementary figure S5. We have also briefly mentioned this in the methods section on page 10, line 362.

As a final purification step, we perform reverse-phase HPLC, which removes all endotoxins (as routinely checked using a limulus amoebocyte lysate assay).

Round 2

Reviewer 1 Report

All my concerns were addressed.

Reviewer 2 Report

The authors addressed the described concerns acceptably.